

**Conjugate Aurora Observations by the Gjøa and Discovery**
**Expeditions**
by
**Alv Egeland**
**University of Oslo, Norway.**
Corresponding author; Alv Egeland. E-mail:  alv.egeland@fys.uio.no
**ABSTRACT**
During 1901 to 1912 - known as the 'heroic period' of Arctic and Antarctic exploration, great
inroads were made not only geographic but also scientific to our knowledge of the continent.
At Amundsen's Expedition through the Northwest Passage measurements of the geomagnetic
field and visual auroras were carried out for 19 months at Gjøahavn (geographic coordinates
68° 37' 10'' North (N); 95° 53' 25''West (W).  Scott's *Discovery Expedition - at Cape*
*Armitage, McMurdo* (coordinates 77.86° S; 166.69° E), Antarctica, carried out same type of
measurements.  Their observations were carried out geomagnetically conjugate to Gjøahavn.
In addition, measurements were overlapping in time during the year 1903-04.  However, these
two stations are located at different longitudes so there is a difference in local time between
the stations of about 6 hours.  Gjøahavn and Cape Armitage are conveniently located for
separating disturbances in the polar cap regions caused by solar electromagnetic radiations or
solar wind.
The observations were carried out for seven moths per year.  This gave a unique possibility to
compare conjugate characteristics of polar cap auroras.  Comparing conjugate geophysical
data introduce some difficulties.  During the winter season at Gjøahavn, they had bright
summer in Antarctica, and vis versa.  Thus, simultaneous temporal, and spatial ionospheric
variations can be marked different.  Still, the diurnal and seasonal variations were similar. The
quantity of the data from Cape Armitage was larger because there they had continuous watch
of the sky.
The main findings regarding polar cap auroras are:
Low intensity bands - also called streamers, are the dominating f orm. The number of events
in 1903 was nearly twice that in 1902 and 1904. A marked midwinter maximum was observed
at both stations. Many displays were observed poleward of the oval. A large fraction was
associated with weak magnetic disturbances.
The polar cap auroral forms: Theta arcs, poleward moving substorm arcs (PSA), and
transpolar arcs (TA), have special geomagnetic signatures, so they can be mapped even if they
are not observed visual.  According to recent satellite measurements they are probably caused
by polar rain and/or photoelectrons.

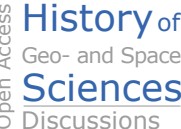

**1.0 Introduction**

During the first decade of the 1900s – known as the 'heroic era' of Arctic and Antarctic exploration, new inroads were made to the continent (cf. e.g. Huntford, 1982; Barraclough and Malin, 1981; Silverman et al., 1994; Egeland and Deehr, 2014). Roald Amundsen (1872 - 1928), and Robert F. Scoot (1868 – 1912) led expeditions, pioneering geological, glaciological, and meteorological discovery. During Amundsen's *Gjøa* Expedition through the Northwest Passage, measurements of the Earth's magnetic field and auroral were carried out at Gjøahavn (GH) on King William's land (geographic coordinates 68º 37' 10'' North (N); 95º 53' 25''West (W), for 19 months. The data have now – for first time, been analysed based on what we have learned during the space age.

During this work it was discovered that Scott's *Discovery Expedition to Cape Armitage (CA), McMurdo* (coordinates 77.86º S; 166.69º E), in Antarctica, carried out the same type of measurements, and that there was overlapping in time with those at *GH* in 1903-04. New calculations showed that Cape Armitage was nearly geomagnetically conjugate to Gjøahavn. This gave us a unique possibility to compare conjugate polar cap auroras.

**1.1 The *Gjøa and Discovery* Expeditions**

Amundsen's and Scott's knowledge of geomagnetism and auroral physics was limited, but Amundsen at least lay the groundwork for serious scientific observations when preparing the *e*xpedition. His main mentor was the Deputy-Director of the Norwegian Meteorological Institute in Oslo, Dr. Axel S. Steen and he also met with Professor Kr. Birkeland (1867-1817). In addition, he cooperated with two German experts, Professor Georg von Neumayer in Hamburg, Director of Deutsche Seewarte Institute, and Professor Adolf Schmidt at Potsdam Observatory. Together with one of his crew they made three trips to Hamburg and Potsdam, Germany. He also travelled to Birkeland's observatory at Bossekopp, in Northern Norway, to learn about geophysical observations. His diary from this Expedition (Kløver, 2017a; Egeland and Deehr, 2014) contains some interesting auroral observations and comments regarding the connections between auroras and geomagnetic disturbances.

Scott's science background and interests for space physics was less. As far as we have found, he never looked on the data. Scientists at the Royal Academy in London were responsible both for the field measurements and the first preliminary presentation (cf. Chree, 1909; Bernacchi, 1908).

As Figure 1 shows, Scott's *Discovery Expedition at CA* was located conjugate to GH, but there is a difference in local time between the station of about 6.5 hours. Because of this location disturbances caused by solar electromagnetic radiations or the solar wind can be separated.


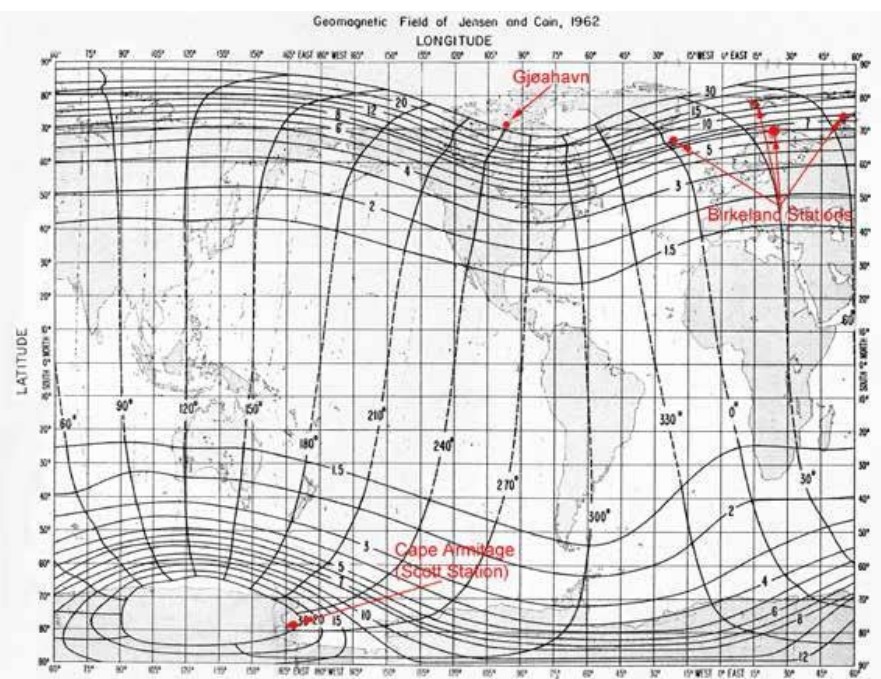

*Figure 1. The figure shows the Earth's magnetic field coordinates at the surface*
*superimposed on a map of the world in rectilinear geographic coordinates. The magnetic*
*longitude – the numbers with the degree mark, is given in degrees east from the point*
*where it intersects the Greenwich meridian and the geographic equator. The magnetic*
*latitude close to the poles is also given in «L- values». These are integer numbers in Earth*
*radii of the distance from the center of the Earth to the point where the field line intercepts*
*the geomagnetic equator (Egeland and Deehr, 2014).*
For interpretation of auroral observations it was discovered – around 1950, that it was an
advance to use a coordinate system based on the geomagnetic field and magnetic time. Such
a system is briefly presented.
The Earth's magnetic latitudes and longitudes are referred to a coordinate system with the z-
axis along the geomagnetic dipole axis through both magnetic poles, which is inclined 11
degrees from the geographic pole in the north hemisphere. and the 180-degree longitude –
also called the geomagnetic meridian, goes through both the geomagnetic and the geographic
north pole. The latitude circles from 0 – 90º north (N) and south (S) are perpendicular on the
geomagnetic meridians.
Local solar time (LT) is defined by the location of the geographic longitude of the station with
respect to the Sun, while universal time (UT) is referred to a geographic system with the 0-
meridian through Greenwich. At local noon and midnight UT the longitude of the observer is
aligned with the Sun and the geographic 0º- and 180º meridian through the geographic pole.
We here compare observations from stations in opposite hemispheres. So rather than consider
the station in the Southern Hemisphere, as 11 hours ahead of Greenwich (UT), we refer to this
station as 13 hours behind in UT and 6.5 hours behind GH, which itself is 6.5 hours behind in
UT.





The intensity and the direction of the Earth's magnetic field is continuous, but slowly
changing.  Based on a new geomagnetic refence field (1969.75 IRGF), the geographic and
geomagnetic coordinates for years 1905 and 1970 for the two stations, have been recalculated.
These calculations show that the magnetic coordinates of CA and GH have changed less than
2 degrees west in longitude (10 minutes) and 0.25 degree of latitude (equivalent to ~25 km)
relative to one another from 1900 to 1970.

| Station | Time | Mag. Noon | Mag. MdNt | SS Onset | SolarNoon | Solar MdNt |
|---|---|---|---|---|---|---|
| **Gjøahavn** | UT | 19:30 | 07:30 | 06:00 | 18:24 | 06:24 |
| | Local Solar | 13:00 | 01:00 | 23:30 | 12:00 | 00:00 |
| **Cape Armitage** | UT | 19:00* | 07:00* | 05:30 | 23:06 | 11:06 |
| | Local Solar | 08:00 | 22:00 | 18:30 | 12:00 | 00:00 |

*The station Cape Armitage may be 15 to 60 min east of Gjøahavn in magnetic time.*
*Table 1.  A list of UT, Local Solar Times, Magnetic Midnight and Magnetic Noon, and*
*Average SS Onset for the two Observatories.*

Magnetic midnight and noon are the local times when the station passes through the plane
containing the Sun and the geomagnetic pole at night and day, respectively.  Thus, magnetic
midnight occurs near 07:30 UT and at 01:00 local time (LT).  The most dynamical changes
and poleward expansion in the auroral zone normally occurs between 22:00 MLT and
magnetic midnight.

**2.0 General about Aurora**
Around 1900, the study of aurora was still an emerging science.  The main question at that
time was the relationship of the aurora and magnetic disturbances (Birkeland, 1908).
In his lecture to the Norsk Geografisk Selskap (The Norwegian Geographical Society), on 25
November 1901, Amundsen (1902) presented his plan for '*The Voyage Through the North*
*West Passage*' (Kløver, 2017a; 2017b).
Amundsen had read Sophus Tromholt's (1885) book, *Under the Rays of the Northern Lights*
and had visit Birkeland in his famous Terrella laboratory at the University.  Nansen's (1897)
drawings of northern lights from the *Fram* Expedition were well known.  Auroral
observations at those high latitudes were unprecedented.
Aurora around the 19-century was only subjective to observations with your naked
eyes - i.e., visual observations.  The pragmatist Amundsen included auroral observations
before bedtime in his daily station activities.
Some basic new auroral facts learned during the space age, will briefly be mentioned.
Spacecraft - after 1960, gave us the opportunity to explore space between the Earth and the
Sun with in-situ observations.  With Explorer 1, launched in 1961, the first measurements
across the near-earth space – called the magnetosphere, were carried out.  Interplanetary
space, not long ago believed to be empty of matter, was filled with streaming electrons and
ions of solar origin.  These streaming particles, called the solar wind, were for the first time
observed during the 1960s.  The solar wind is the important connecting link between solar
activity and geophysical disturbances.  The interplanetary magnetic field (IMF) is an

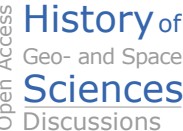
extension of the solar magnetic field carried by the solar wind as the plasma leaves the sun
(Kivelson and Russell, 1995).

**3.0 The Auroral Observations at Gjøahavn**

The map in Figure 3 illustrates the location of the GH Observatory relative to both the
geographic and the magnetic pole, the Artic Circle as well as the location of the new auroral
station at Spitzbergen.
Aurora occurrence in the 19th century was normally referred to the Fritz' (1881)
auroral zone, with maximum at 67 degrees magnetic latitude - or 23 latitude degrees from the
magnetic poles.  The Gjøahavn station is located poleward of this zone.
Observations were carried out from the end of September to mid-March by Peter Ristvedt
(Kløver, 2017b).  Ristvedt's original auroral handwritten notebook is not easy to read, but
Aage Graarud (1932) translated his notes into English, word for word.  The first page in
Graarud's version – out of four, is shown in Fig. 2.  Amundsen reported 15 events in his diary
(Kløver. 2017a) in addition to those listed by Ristvedt.
No official classification of different auroral forms existed then.  The first aurora atlas was not
published before 1930, by Carl Størmer (1930; 1955).  All data from this expedition are stored
by Videnskabs-Selskabets Skrifter, No. 3 (cf. Norwegian Geophysical Committee, 1920).
Following terms were used in the auroral protocol:
*Streamers/strips*, which have been taken to mean active auroral rays.
*Band(s)*; seem to be a common name for both arcs and bands.
*Crown*; is interpreted as corona.
*Auroral clouds*; large surfaces of lights.
*Auroral patches*; this form is not defined, but is probably like clouds, but smaller.
Words as '*auroral fire, flaming auroras, flickering streamers of lights* are also listed in
the original protocol.

### Observations of Aurora Borealis.

| Years | Days | Hours | | Aurora Borealis. |
|---|---|---|---|---|
| 1903 | November 4 | 5 30 p.m. | ........... | Streamers from SE to zenith. |
| » » | » 8 | 6 » | ........... | Green, magnificent bands from SE to zenith. |
| » » | » 10 | 5 30 » | — 6 20 p.m. | Faint streamers and bands, frequently and briskly changing. |
| » | » 11 | 5 20 » | — 6 30 » | Faint streamers, S—SW. |
| » | » 12 | 3 a.m. | ........... | Faint streamers and bands, SE, SW, hor. to zenith. |
| » | » 12 | 6 p.m. | ........... | Faint horizontal stripe at the south horizon. |
| » | » 12 | 11 » | ........... | Faint streamer in the south. |
| » | » 12 | 12 » | ........... | Faint stripe S to W ca. 30° above the horizon. Thick mass of clouds at the base. |
| » | » 13 | 12 » | ........... | Very faint streamers from the horizon, ca. 45° towards the zenith. |
| » | » 14 | 10 » | ........... | Cloudlike aurora in S and W, ca. 30° above the south horizon, ca. 10° above the W-horizon. |
| » | » 14 | 11 » | ........... | Aurora in S and SW, 20° above the horizon with dark clouds underneath, a single streamer 60° towards the zenith. |
| » | » 14 | 12 » | ........... | Bright stripe SE — N, at the highest 30° above the horizon. |
| » | » 17 | 8 20 » | ........... | When going from ship, we saw a luminous beam above our lodge, and believed that this was on fire, but reaching the top of the hill we saw the light to be an aurora in the S. It had the shape of a large fire on the ice between us and the horizon. Gradually the streamer lengthened along the ice as far as to the W. Then flickering streamers commenced to stretch towards the zenith. At 6ʰ 15ᵐ p.m. growing fog hindered further observation. |
| » | » 17 | 10 » | ........... | Faint aurora in W, vague in form. |
| » | » 17 | 11 » | ........... | Very faint aurora in W. |
| » | » 17 | 12 » | ........... | Widely spreading aurora, but still faint, with its centre at the W-horizon, from there sending streamers towards the zenith. In the northern sky a faint aurora of vague form. Only in S the sky was clear. |
| » | » 20 | 8 30 » | — 9 p.m. | Strong belt from the S hor. towards SW, altitude ca. 20°. |
| » | » 22 | 5 30 » | ........... | Faint stripe from S to W horizon. |
| » | » 23 | 6 » | ........... | Faint aurora as streamers from the S and SW hor. towards the zenith. Ca. 45° high. |
| » | December 8 | 9 » | ........... | Strong flashing zigzag beams from the SW hor. through the zenith to the E horizon. |
| » | » 8 | 11 » | ........... | Faint streamers in S. |
| » | » 9 | 3 30 » | ........... | Faint, bright streamers from the S hor. towards the zenith, ca. 30° high. |
| » | » 10 | 5 30 » | ........... | Aurora as a faintly flickering flare in the SW horizon. |
| » | » 15 | 5 » | — 9 p.m. | Arch S—W in SW ca. 15° high. |
| » | » 17 | 4 30 » | ........... | Very strong aurora as streamers SW-NE through zenith. |
| » | » 17 | 5 30 » | ........... | Still strong streamers, but on the zenith several larger and smaller spots in lively motion. |
| » | » 17 | 9 » | ........... | Very faint, hardly visible pavilion in the zenith. |
| » | » 18 | 5 » | — 9 p.m. | Stripe SW-WNW. Some faint and frequently shifting points at zenith. |
| » | » 21 | 5 30 » | — 8 40 » | Bands through zenith all around the horizon. |
| » | » 24 | 1 a.m. | ........... | Very strong band S-WNW ca. 30° above the horizon. From SW a streamer to zenith, fading away towards the zenith, where it wholly disappeared. |
| » | » 26 | 11 30 p.m. | ........... | Faint streamer from N to zenith, where it entirely went out. |
| » | » 30 | 8 a.m. | ........... | Strong zigzag band from N hor. to zenith, frequently changing to flickering streamers that rapidly died away. |
| 1904 | January 5 | 5 p.m. | — 6 p.m. | At hor. S-W. On the SW a faint streamer to the zenith. |
| » | » 5 | 10 » | ........... | Strong flickering streamers SE-WNW, stretching from the hor. to zenith, there forming a pavilion (aur. corona). |
| » | » 6 | 4 30 » | — 9 » | Bands and zigzag streamers at the hor. ESE-NW stretching to zenith, rapidly shifting in power and colour. Greatest intensity between 5ʰ and 5ʰ 30ᵐ. |



- *Figure 2. One of the pages of Graarud's (1932) collection is shown. Important*
- *events – year, dates, and times, are briefly commented on in the text.*

Amundsen shows scientific effort for the observing program, but few recordings are included
in his book *The Northwest Passage* (Amundsen, 1908), except on the drift of the magnetic
north pole since Ross (1834) measurements in 1831. That is further summarised in his lecture
at the *Royal Geographic Society*, London on 11 February 1907 (Amundsen, 1907).
However, Amundsen appointed a committee consisting of Dr. Aksel S. Steen, Deputy
Director at The Norwegian Meteorological Institute, as chairman, while Dr. Wasserfall and

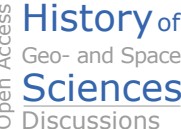
the meteorologist N. Russeltvedt were the other two members. In fact, the editing and
preparation of the observations were not complete until 1933 (Steen, Russeltvedt, and
Wasserfall, 1933).
Northern lights poleward of the auroral zone are called polar cap auroras. Hardly any
documentation of this type of auroras existed when these expeditions were carried out. As a
clever expedition leader, Captain Amundsen knew the value of keeping day-to-day records.
However, the instructions indicate that visual monitoring of aurora did not have the highest
priority. Fortunately, the detailed diaries of the crew members have now been published
(Kløver, 2009, 2017a; 2017b, 2017c).
The dataset has limitations because: 1) observations were not carried out around the
clock, 2) no illustrations or sketches exist, and 3) the available descriptions are scanty.
During the epic voyage through the rest of the Northwest Passage to Alaska, they also
observed some displays, but they are not included here.

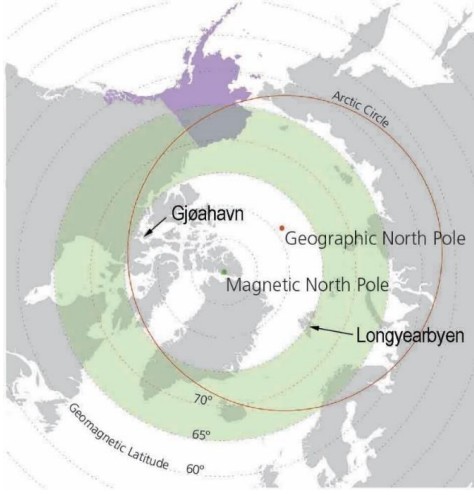

*Figure 3. A map of the Arctic region where the locations of Gjøahavn, the magnetic and the*
*geographic poles are marked.*

**3.1 Scientific Results of the Gjøahavn Auroral Observations**
The text for the first event on the 4th of November, 1903 is: *Streamers of northern lights from*
*southeast (SE) to the zenith (Z), were seen in the early local evening (from 5 pm.).* In
Amundsen's diary (p. 124), it is listed that the temp. was - 25º C, and he saw *'northern lights*
*from early afternoon. They appear as a semi-circular formation approximately 30º above the*
*southern horizon. 5 rays stretched toward the zenith from the semi-circular formation. The*
*rays came and disappeared, intermittently. These lasted about a quarter of an hour and then*
*disappeared. I also saw a fan-shaped clouds to the NW.* The magnetic record for the 4th to
the 5th of November, is shown in Figure 5.
This event is illustrated in Figure 4. The regular oval auroral oval - 30º above the
southern horizon, together with three arcs stretching poleward from the oval, are shown.

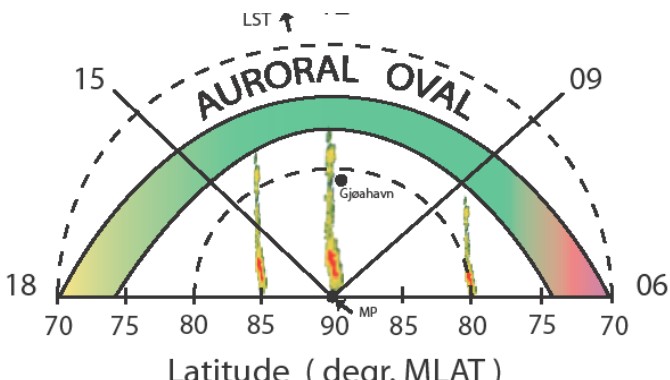



**Fig. 4.** *This is a schematic illustration of the aurora on 4th November 1903 at GH, in*
*magnetic time (MLT) and magnetic latitude (MLAT), covering the region from the magnetic*
*pole (MP) to 70 degrees north. The viewing perspective is the polar upper atmosphere with*
*the oval – in yellow green, from 06, via noon to 18 MLT. The direction to the Sun is up in the*
*figure. The location of GH is marked. Three reddish sun-aligned-polar arcs stretching from*
*the oval past the zenith, are shown.*

Thus, this auroral event started before magnetic noon. When the intensity of the lights
changed rapidly and moved poleward, the largest magnetic disturbances were recorded, as
shown in Fig. 5.

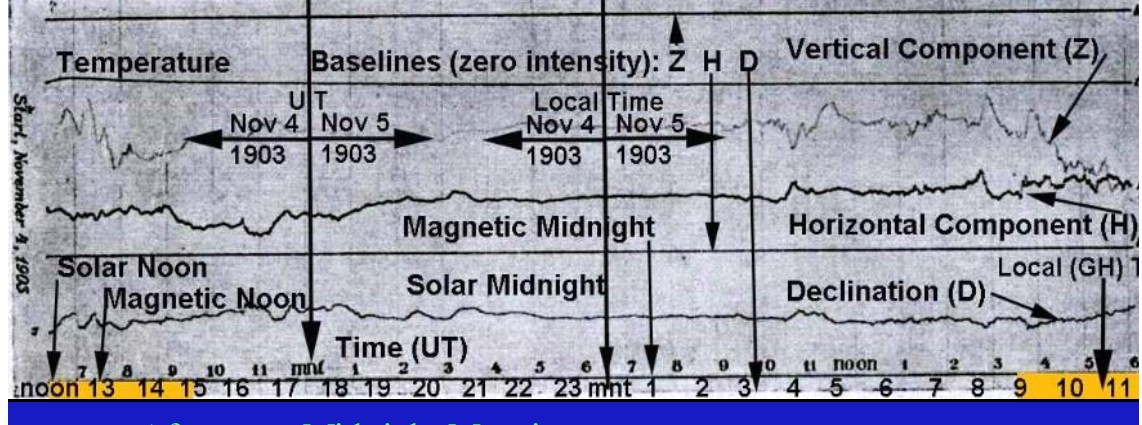



Figure 5. *The magnetic field recordings - as function of time in UT, from the 4*[th] *to the*
*5. November 1903. The three curves - marked Z, H and D, illustrate the vertical, the*
*horizontal and the declination components of the magnetic field. The upper curve shows the*
*temperature in the recoding hut and the baseline, while the bottom curve gives the hours. The*
*vertical scale illustrates the magnetic intensity in nT. The photosensitive paper used for the*
*recordings was changed daily near 6 UT, i.e. near magnetic noon. Notice that the magnetic*
*activity is very low around magnetic midnight.*

The mistake of identifying an auroral event as a fire, indicates that these *luminous*
*beams* had a marked reddish colour.  The emperor Tiberius, in the year 37 A.D., ordered a
troop of solders to rescue the village of Ostia which was reported to be on fire.  It was an
unusual fire seen from large distance, which was 'an aurorl fire'.  There are a few more recent
examples of "auroral fires".  The one reported from London on the 15th of September 1839,
when the whole sky was one vast sheet of reddish light, is well documented.  It had a most
alarming appearance.
An 'auroral fire' was also seen at GH.  According to Amundsen – p. 134 in the diary,
the northern lights on the 14th November lasted passed midnight.  The event was observed
by the crew who were out reindeers-huntingthe temperature was – 20º C.  On their return
they saw a big reddish display in the sky and were afraid the whole camp was on fire.  Quote:
"We saw the luminous beam above the lodge, and believed that the camp was on fire, but
reaching the top of the hill, we discovered the light to be aurora.  It had the shape of a large
fire on the ice between us and the horizon.  Near midnight aurora was sending up streamers
toward zenith".  Northern lights were also observed on the 18th, 20th as well as on 22nd and
23rd November, but without magnetic activity.
During the first Christmas Eve '*a strong band of northern lights ~ 30º above the southern*
*horizon';* was seen.  In addition, streamers of lights reaching the zenith, were observed.  This
is an event like the one mentioned on the 4th of November.  The activity lasted from early
afternoon to after local midnight.  Amundsen was surprised - p.151 in diary, *as* he wrote: *The*
*strong aurora I saw at Christmas, was almost with no geomagnetic disturbances*. Regarding
the auroral event he wrote: *Very strong aurora as streamers SW- NE though zenith is seldom.*
*Later in the afternoon, strong auroras were seen near zenith.  Several larger and smaller*
*spots in lively motion with hardly any magnetic disturbances*.
In the diary on page 172, Amundsen wrote: *We saw active northern lights over the*
*whole sky on the 8th of February and it lasted three days.  They seem to come from all*
*directions.  Around midnight the lights spread over large part of the sky, and faint auroras*
*were even seen all up to the north.  No auroras were seen in the south.  Strongest auroras in*
*the western horizon with poleward moving up to zenith, are mentioned.  They appear one*
*moment and disappeared the next.*  As the rays were deep reddish, it may indicate an
observation of Sun-aligned-arcs.  The arcs were most clearly seen on the darker eastern part
of the sky.  These observations were carried out two hours before magnetic midnight.  The
day after Amundsen wrote*: 'The northern lights I observed yesterday have slightly disturbed*
*magnetic activity.*
February 1905 was also an active period with auroras five days in row.  For one event the text
is: *Aurora in SE to NW, brisk motions, sometimes dispersed over the entire sky and with*
*much deeper colours*.
The last observation from Gjøahavn is on 2nd March 1905.  *Aurora is seen all over the*
*entire sky from a strongly bright pavilion in zenith.*
Total numbers of nights per months when auroras were observed, are shown in Table
2.
90 events were recorded.  The overwhelming number of events were of moderate or low
intensity.  26 out of 90 events extended zenith.  Only on 7 of the 90 events were colour listed.
Most dynamical changes and poleward expansions of auroras normally occurred between
22hr MLT and magnetic midnight.  Polar cap auroras were visible from magnetic noon in
December, i.e. from 1300 LT, but few observations were listed before after 1500 LT.

| Year/ month | Sept | Oct | Nov | Dec | Jan | Feb | Mar |
|---|---|---|---|---|---|---|---|


| 1903 30 | 1 | 3 | 13 | 13 | - | - | - |
|---|---|---|---|---|---|---|---|
| 1904 45 | 0 | 5 | 12 | 8 | 11 | 6 | 3 |
| 1905 15 | - | - | - | - | 8 | 5 | 2 |
| Totals 90 | 1 | 8 | 25 | 21 | 19 | 11 | 5 |

*Table 2. Total number of auroras per months, observed at Gjøahavn. The events reported in*
*the diaries by all crewmembers have been included. Thus, this table is different from the one*
*by Graarud (1932).*

**4.0 Auroral Observations at Cape Armitage**
The observations were led by L. C. Bernacchi (1908), but carried out around the clock by 'the
meteorologist on duty with a check every hour'. Dr. Bernacchi was called up when
significant, large auroral sightings were observed. Even if the observers were at outlook for
24 hours per day, the conclusion is 'still some faint or moderate bright auroras might have
failed to be noted'. A similar auroral classification as at GH is used. Regarding auroral
intensity, the conclusion is: Their brilliances were rarely more intense than stars of the 4th
magnitude or the Milky Way.
Luminous patches - sometime small and at other times occupying almost the whole sky,
which frequently looks like the appearance of clouds, are mentioned.
Streamers are often listed and can represent different forms. Vertical rays close together is
mostly likely what Størmer (1930) called *draperies*. Spectroscopic observations were tried,
but not successful due to the low intensity of the instrument, even if long time integration was
tried.
Arcs and bands touching the horizon at both ends were rarely seen. The aurorae were
particularly visible during the dark moon periods. During exceptional extensive displays,
Bernacchi called Mr. Edward A. Wilson, the expedition junior surgeon who also was an artist,
to clearly see the largest displays. Wilson contributed with two dozen charcoal drawings of
aurora australis. From these sorties, one is shown in Figure 7.

TABLE showing Number of Days in each Month when Auroræ were Recorded.

| Year. | March. | April. | May. | June. | July. | August. | September. | Total. |
|---|---|---|---|---|---|---|---|---|
| 1902 | 0 | 10 | 8 | 11 | 10 | 9 | 4 | 52 |
| 1903 | 2 | 18 | 14 | 18 | 22 | 14 | 2 | 90 |
| Days. . . | 2 | 28 | 22 | 29 | 32 | 23 | 6 | 142 |

*Table 3. The recorded auroral events for the different months in 1902 and 1903 are listed.*
*As the table shows aurore were recorded between March and September- particular in 1903,*
*but the main activity was found both years during June and July (Bernacchi, 1908).*
*Surprisingly, few auroral events are observed in May both years.*

The numbers of auroral events observed each month in 1902 and 1903 are listed in Table 3.
The number is significant higher – nearly double, in 1903 compared to 1902. Maximum
occurrence was recorded both years mid-winter. The activity is very high in April both years,





while the numbers of days with auroras are nearly equal in both May and August. It is
interesting to learn that more events were even seen in April than in May. This confirms well
with what was found at GH namely, that aurora is a very dynamic phenomenon. The year
1903 was special with the strongest storms during that century. It is also interesting to notice
that a similar series of storms - called the Halloween storms, were also recorded 100 years
later.
**4.1 Scientific Results of the Auroral Observations at GH and CA**
During the space age auroral observations have been carried out continuous and the statistical
locations of the auroral ovals have been established. Its location – both in north and sought,
for moderate disturbed conditions – $K_p$ =2, is shown in Figure 6. The red sector of the oval
illustrates when daytime auroras dominate.

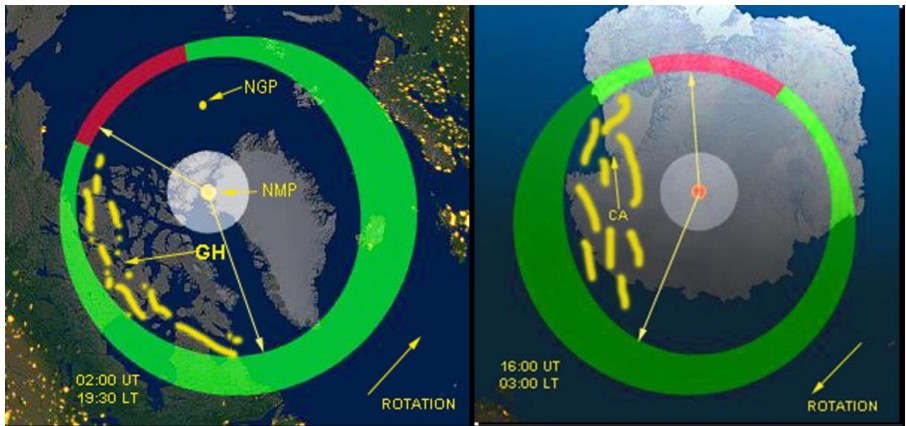

*Figure 6. This figure illustrates the location of the Arctic and Antarctic auroral oval during*
*moderate disturbed magnetic condition [Kp at both stations around 2] according to*
*Breedveld (2020) and Sigernes et al. (2011). The locations of the stations are marked.*
*Approximately 30 % of the events are observed poleward of the oval. The rotation direction –*
*marked by the yellow arrows, is opposite in the two hemispheres. The figure is based mainly*
*on ground observations. Low intensity auroral yellow bands in the* geographical *western*
*hemisphere – called 'Sun-aligned Arcs', were the dominating auroral form poleward of the*
*oval at both GH and CA.*
The accurate location and extend of the oval depend on many scientific processes and is a
very dynamical region. The measurements at GH and CA clearly show maximum occurrence
during midwinters and its connection with magnetic disturbances is low, which show that our
data are observed poleward of the oval indicate that the production of polar cap auroras is
somewhat different from oval auroras.
One of the charcoal auroral drawings from CA is presented in Figure 7. To the author it looks
like a beautiful folding curtain. Edward A. Wilson was the artist who made the drawing.



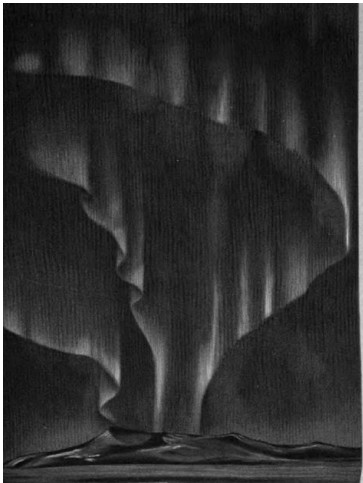

*Figure 7.  A charcoal rendering of the aurora observed at CA on April 9th, 1902, at 0225 am*
*LT of an event classified as streamers from north with intensity as a star of magnitude*
*3.  Its title "Auroral Streamers". Copied from Bernacchi's (1908).*
**5.0. Daily variations in the occurrence of auroras at CA and GH**
The diurnal variation of auroral occurrence at CA for the two winters 1902 and 1903 is shown
in Figure 8.  The average sightings reveals that a large fraction was observed as poleward
expansions from the auroral zone.  Weather and moonlight tend to decrease the number of
dayside auroras.  According to Bernacchi (1908), 'many of the observations were made when
the magnetic curves were quiet, or even very quiet'.

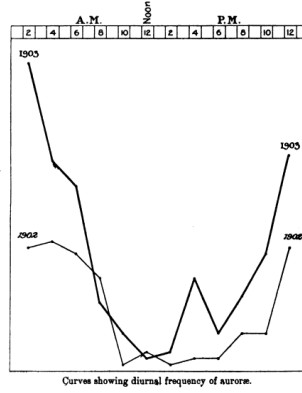

*Figure 8.  The diurnal variations of auroral occurrence versus local time are shown for the*
*winters 1902 and 1903 (*Bernacchi, 1908)*. Maximum was observed both years between 10hr*
*pm. to 04 am.  The curve for 1903, clearly illustrates a second maximum during early*
*afternoon.  The peak around 3 and 4 local time overlaps with low energy electron*
*precipitation observed by auroral rockets.  The number of events is listed in Table 3.*

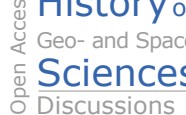

The time of occurrence appeared to depend upon the latitude where the displays were
observed.  If the events happened closer to the pole, it usually started to appear earlier.
Maximum was first observed in the region which first got dark.  Several events north of the
auroral zone were observed early afternoon.  During five months it was too much daylight for
auroral observations.  That many more events were observed in 1903 than in 1902 is
explained by the fact that the number of sunspots was higher in 1903; namely 53 compared to
24 spots in 1902 (Chree, 1909; Egeland and Deehr,2014).
Regarding interpretation of the observations, Chree (1909) discussed electric currents in the
atmosphere in the following way: 'The existence of very bright auroral band or streamers may
mean an electric current of unusually high intensity contribute'.
The average diurnal variations at both stations is shown in Figure 9.  The data from GH are in
blue while the CA measurements are in red, The number of auroral days is given by the left
vertical scale.  Local afternoon (Aft), magnetic noon (Noon), and magnetic midnight (Midnt),
are marked.  The bottom yellow, horizontal line illustrate when observations were not
possible at CA – because of too much sunlight, while the blue, horizontal line illustrate that
systematic auroral observations at GH were not carried out after 22 LT.  The two stations are
6 hours separated in local time, and magnetic time rotates in opposite directions in the two
hemispheres.
The diurnal variations follow closely local solar time with a marked maximum near local
midnight.  The GH data also show high activity early afternoon – when we observe the sun-
aligned arcs.  Aurora around magnetic noon – called daytime auroras, were observed.  At GH
it was too much sunlight near magnetic noon, while few events were recorded around
magnetic midnight.  The observations indicate that the polar cap auroras - with maximum
occurrence during midwinters, contribute significant to the total occurrence.  In addition, the
relations between auroral occurrence and magnetic time is different from what has been
observed in the auroral zone ( cf. e.g.  Sandholt et al., 2005).

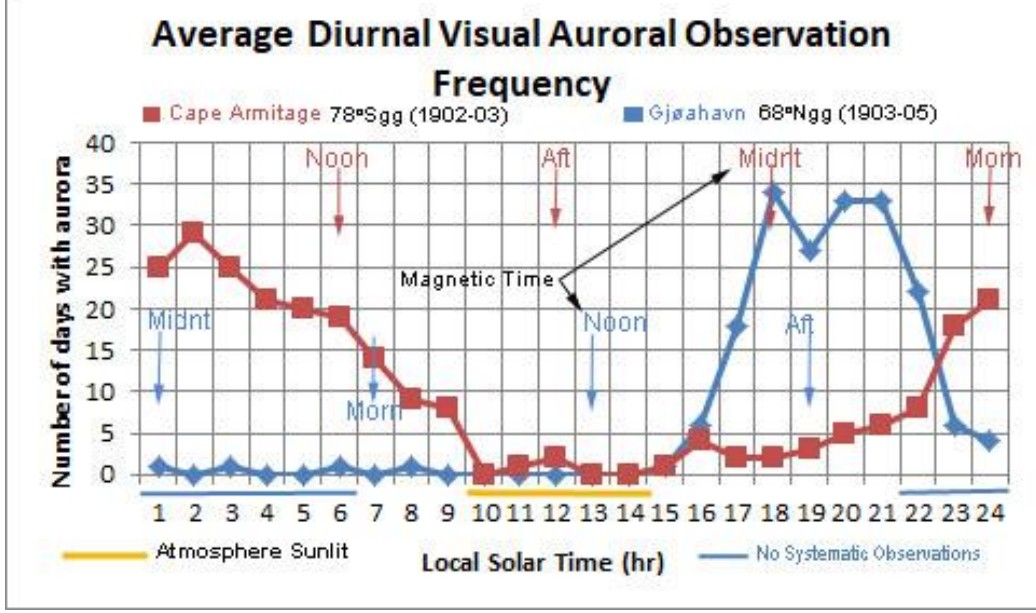


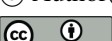




*Fig. 9. The average diurnal variations in auroral occurrence at GH in blue curve, and at CA*
*in red, as function of local solar time, is shown. The number of days with auroras is given by*
*the left vertical scale. The time for local afternoon, magnetic noon, and magnetic midnight,*
*are marked. The bottom yellow, horizontal line illustrate when observations were not*
*possible at CA – because of too much sunlight, while the blue, horizontal line illustrates that*
*no systematic observations at GH were carried out between 22 and 12 LT. The stations are 6*
*hours separated in local and magnetic time (from Egeland and Deehr, 2024 in the Fram*
*project, not published yet).*

   .

The correlation with magnetic disturbances is low, different from than found for oval auroras.
Maximum occurrence is found several hours after magnetic midnight. Thus, *these* data
indicate that the diurnal variation is not controlled by the solar wind.

**6.0 Theta polar arcs (TPA), Polar Substorm Arcs (PSA), and Transpolar Arcs (TA), are**
**the dominating auroral forms observed.**

      ,
      **6.1 Theta polar cap arcs**
The form called *theta* aurora was observed in the 1980's when we got satellite photographs of
the entire polar sky. The name was chosen because - with an arc stretching from one side of
the oval to the other, it is like the Greek letter theta (cf. Fig. 10). Thus, the arc has a 'noon to
midnight' alignment. Because its intensity is low, this auroral form is not often seen by the
naked eye. The arc across the polar cap has a definite orientation in the sun-earth direction.
They are primarily excited by low energy - < 100 eV, electrons. Excite oxygen atoms above
200 km, yielding weak reddish aurora and virtually no emissions below. This form of aurora
was generally not associated with magnetic disturbances on the ground.
      Theta polar cap arcs were observed at boat station, but from a ground station only a
small part of the theta form is seen. A large part of the Sun-aligned arc is stretching outside
the field of view of the observer.

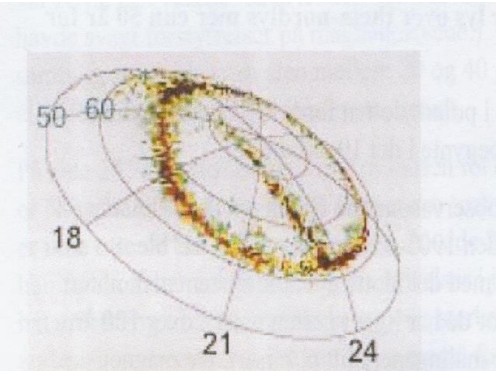

**Fig. 10.** *Picture of theta*
*aurora over the northern*
*hemisphere taken by the IMAGE*
*satellite – in 2002. The sun is in the*
*upper part of the picture. (Photo;*
*NASA).*

-

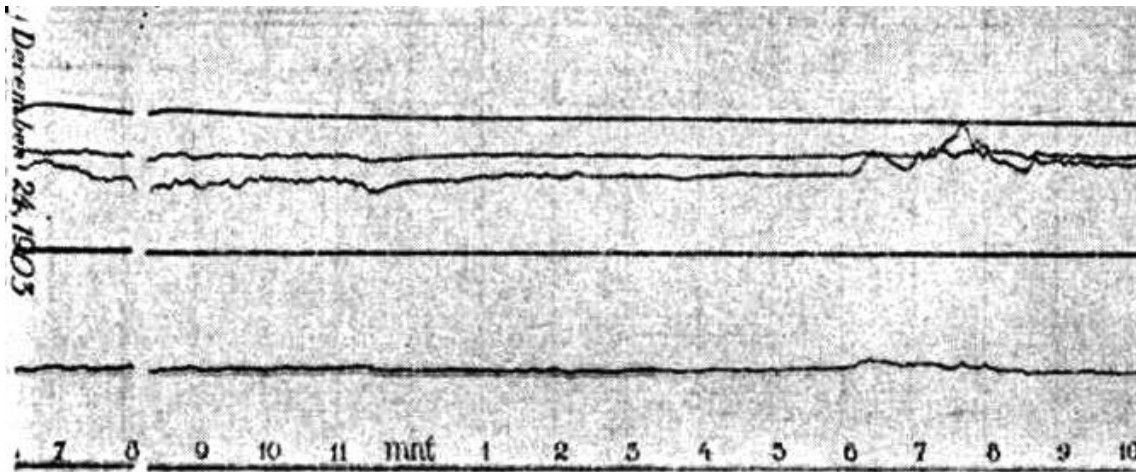


*Figure 11. Even if the magnetic recording at GH shows low magnetic activity during theta*
*auroral activity, some weak disturbances are recorded in the vertical component.*

**6.2 Poleward moving substorm arcs (PSA)**
On the 14[th] of December near 2230 magnetic local time (see Figure 12), we observed a
poleward expansion associated with the onset of an auroral substorm – both at GH and CA,
one hour before magnetic midnight.
Similar magnetic effects of poleward expansion of aurora were observed simultaneously in
both hemispheres – for a few other events, near 22hr MLT.

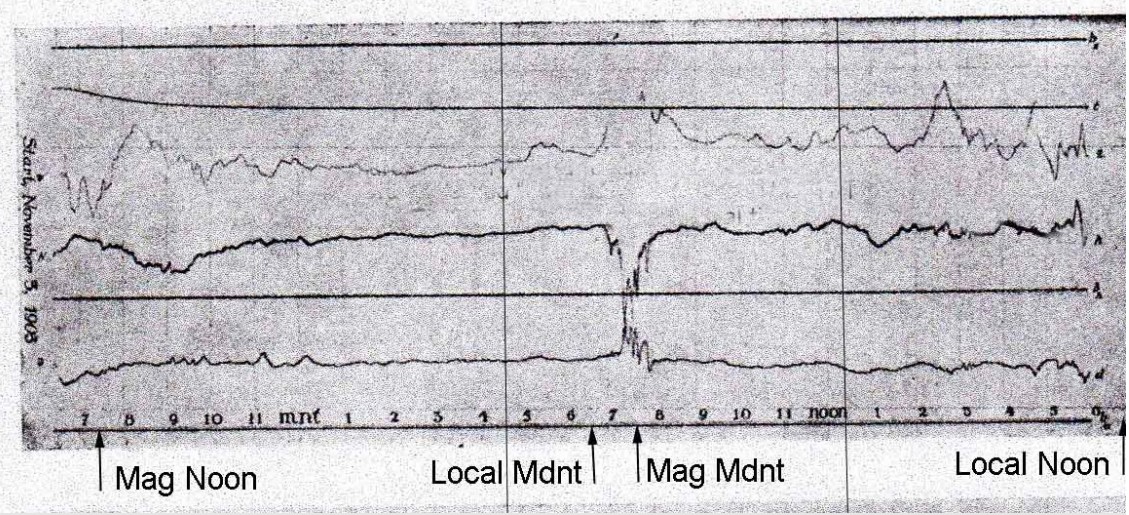



*Figure 12 shows marked disturbances in the three components near magnetic midnight, or at*
*730am in UT. This has been observed at both stations during the occurrence of PSA auroras*
*(Egeland and Deehr, 2024 in the Fram project, not published yet).*
.
**6.3 Transpolar arc (TA)**
Low intensity bands – called Sun-aligned arcs, illustrated in Figure 6 by the yellow strips, are
the overwhelming number of auroral forms observed poleward of the oval.  However, no such
forms are observed +/- 2 hours around local midnight.  Magnetic recording for such events
(see Figure 5) shows that the magnetic activity is extreme low around solar midnight, while
some activity is observed both before and after. An east west transpolar arc illustrated by the
arrow, recorded from above, is shown in Figure 13.

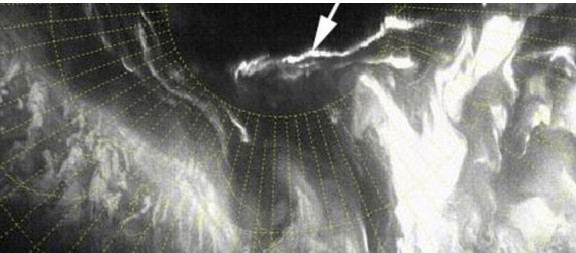

*Figure 13. Example of an east west transpolar auroral arc, north of the oval – marked by an*
*arrow in the picture – taken from a satellite (Egeland and Deehr, 2024 in the Fram project,*
*not published yet).*
**7.0 Summary and Conclusions**
Because of the unique locations geographic and geomagnetic (see Fig. 1), GH and CA share
the same magnetic time, but are separated approximately six hours in geographic time.
During mid-winter conditions at GH, they had midnight sun at CA where the ionospheric
conductivity is significantly higher than during winter.  The geomagnetic field was recorded
with the same type of instruments (cf. Steen et al., 1933; Chree, 1909).  Media from that time
shows that Amundsen enjoyed the advantage of plurality (cf. Amundsen, 1907; 1908; &
1927).  The Discovery data were taken care of by the Royal Society in London, and
preliminary results were published in internal reports nearly twenty years before the GH data
(cf. Chree, 1909; Steen et al., 1933; Egeland and Deehr, 2014).  Their observations have not
received much attention.
At the beginning of the 20[th] century, it was generally concluded that when northern lights
appeared overhead, the earth's magnetic field is disturbed (cf. e.g. Tromholt, 1886;
Birkeland, 1908; Chapman and Bartels, 1940; Størmer, 1955; Chapman, 1968).  Based
on several statements in Amundsen's diary (Kløver, 2017a) such as: "strong auroras,
but no magnetic disturbances", indicate that Amundsen was surprised over such findings.
This suggest there may be other auroral patterns in the polar cap regions than within the
auroral zone.
Height information on these old auroral data does not exist.  The reddish northern lights
which peak at altitudes above 200 km, cause hardly any magnetic disturbances.  So, when
Amundsen – seven times, observed 'reddish northern lights with nearly straight magnetic

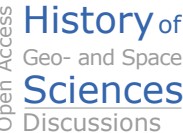

lines', these events most likely occurred above 200 km.  On page 141 in his diary Amundsen
wrote: '*Until now we have had an opportunity to see several times how certain strong*
*northern lights haves no influence whatsoever, but faint lights may cause greater magnetic*
*disturbances'.*  Thus, a  one-to-one correlation, between auroras and magnetic
disturbances, is not always true.
The increase in the horizontal magnetic component – near 0600 UT on the 14[th] December
1903 - when a poleward expansion aurora was observed at GH, most probably occurred
simultaneous at CA.  Similar magnetic effects of poleward expansion of aurora were observed
in both hemispheres, near 22hr MLT.  Unfortunately, we lack statistical data to prove this
finding.
The diurnal and the seasonal variations at both stations have great similarities.  The six
months separation in the regular season radiations mainly influence the day to day variation.
The basic new findings from the auroral observations at CA and GH, are:
•   240 auroral events were observed during the two seasons.
•   Maximum activity occurred near midwinter both in the northern and southern
hemisphere.  The auroral season was nearly seven months per year.  There was too
much daylight for visual observations the other months.
Auroral colours were noted on only a few percents on the events.
Low intensity bands, also called streamers, dominated the occurrence.
The number of events was nearly double in 1903 compared to 1902 and 1904. Thus,
1903 was an
active auroral year.
Many events extended to zenith and some even further poleward.
A large fraction of the observations were associated with some weak magnetic
disturbances.
Three aurora forms, namely: Theta polar arcs, poleward moving substorm arcs, and
transpolar arcs, dominated the polar cap auroras. These three forms have special magnetic
signatures, so they can be mapped even if they are not seen. The main reason that not
more *polar cap arcs* are observed is probably because they are normally subvisual.
According to recent satellite measurements polar cap auroras are caused by polar rain
and/or photoelectrons.

Professors Kr. Birkeland (1867-1917) and C. Størmer (1874-1957) before the space age
mapped the occurrence and characteristics of auras based on ground measurements (cf.
Birkeland 1908; Størmer, 1955). When in-situ recordings started by rockets and satellite, new
auroral forms and processes – such as polar cap auroral substorms, new auroral forms and
dayside auroras, were discovered. Still, coordinated auroral ground tracks are important. The
conclusion accepted for more than a century that when auroras occur overhead, the Earth's
magnetic field is disturbed, must be modified after polar cap investigations have been carried
out. The connections between auroras and magnetic disturbances are more complex than in
the oval. Furthermore, the occurrence and similarities between conjugated polar cap auroras
are more difficult to investigate when one hemisphere is complete dark while midnight sun
dominate the other.



Some weak connections between polar cap auroras and geomagnetic activity are observed.
Based on these measurements it has been investigated if polar cap auroras can be identified
from the magnetic recordings. Establishing a relationship between the various types of polar
cap aurora and the solar wind is hindered by the sporadic nature of visual and optical
observations. Continuous, conjugate geomagnetic records in the north and south may provide
a means of solving this problem. Over the central polar cap, the situation is probably even
more complicated. Further investigations are needed to find out if a different generation
process for aurora occurs when the particle precipitations may not controlled by the solar
wind, but by the electromagnetic radiations from the sun.

## Competing Interests

The contact author has declared that none of the authors has any competing interests.

## Acknowledgements

The author expresses his special warm thanks to dr. Charles Deehr, Fairbanks, Alaska, who
was an active coworker with valuable contributions when this project started. However, after
his later illness, he has withdrawn from this work. I will also like to thank Fram Museum,
Oslo, for some financial support.

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

Tromholt, Sophus, 1885: Under the rays of the Aurora Borealis in the Land of the
Lapps and the Kvaens. Vol. 1 & 2, Houghton, Mifflin, & Co. Boston.
Videnskabs-Selskabets Skrifter, No. 3. Norwegian Geophysical Committee, 1920: Various
Papers on the
Projected Cooperation with R. Amundsen's North Pole Expedition,
Geofys. Publ. 1, No. 4.