# Peer review of "Conjugate Aurora Observations by the Gjøa and Discovery Expeditions"

_History of Geo- and Space Sciences, 2023_

## Community Comment (CC1)

**Review of paper submitted to History of Geo- and Space Sciences**

**Title:** Conjugate Aurora Observations by the Gjøa and Discovery Expeditions
**Author:** Alv Egeland
**MS No.:** hgss-2023-14

**General comment**

The paper is a review of the polar expeditions headed by Amundsen to the Northwest Passage at Gjøahavn in the Arctic and Scott's to Armitage at McMurdo in Antarctica, respectively. Focus is on conjugate auroral observation back in the early 1900. The language is clear, and the paper is interesting to read.

Based on expedition members logbooks and reports, the occurrence of auroras is reported and discussed in relation the magnetic activity / response at ground level for each site. The author concludes that the dominant forms observed must have been mainly substorm, theta and transpolar arcs. The latter two is usually found poleward of the auroral ovals. The only clear reported magnetic variation / response was found the be associated with substorm aurora.

As stated in the summary:

*Thus, a one-to-one correlation, between auroras and magnetic disturbances, is not always true.*

This must have confused the expeditions members, since it is much later on during the Space age that started with in-situ rocket and satellite exploration back in the late 50's, that the complex behavior of magnetic disturbances and aurora forms was mapped.

In addition, it was interesting to learn the history behind the expeditions and all the people involved in planning and conducting the magnetic measurements and observations.

*To conclude, I recommend the paper to be published with minor corrections.*

**Minor corrections**

Page 1 - line 32: remove spacing in word: f orm.
Page 2: lines 42 – 50 is very similar to first paragraph on page 1 (feels like repetition).
Page 9:  line 238: misspelling: 'an aurorl fire"
Page 11: line 331: reference to Sigernes et al. (2011) is not listed in the reference list.

F. Sigernes, M. Dyrland, P. Brekke, S. Chernouss, D.A. Lorentzen, K. Oksavik, and C.S. Deehr, Two methods to forecast auroral displays, Journal of Space Weather and Space Climate (SWSC), Vol. 1, No. 1, A03, https://doi.org/10.1051/swsc/2011003, 2011.

Sincerely

Fred Sigernes                                                                                    UNIS, 30.01.2024

---

## Author Comment (AC1)

Ref. no 3

I thank Ref. 3 for valuable and positive comments. Her follow my answerers and comments to his comments:

1) Meaning of the received results in relation to modern space physics should be described more clearly. Answer: The manuscript has been updated on this point, and regarding the language.

2) English in paper (style, first of all) is unsatisfactory. Answer: Doing the work with a new check of the paper I certainly hope my English has been improved, but I am not an expert.

4) It is recommended changing the paper title as follows: "Conjugate Aurora Observations in Greenland ??? and Antarctic by Amundsen's Expedition through the Northwest Passage and Scott's Discovery Expedition". Answere: This 'Title' is very long.

Will consult editor.  Will not change. The title include the important words rg. This paper

5) The paper is recommended to publication with regard to these remarks. "Thank you"

---

## Author Comment (AC2)

Remarks to referee 2

. I wish to thank Ref. 2 for positive comments, and recommendation for publication.

1) Form has been corrected.

2) The lines 42-50 has been reduced and rewritten.

3) Auroral fire spelled correct.

4) The reference to Signernes et al., 2011 has been put into Reference list.

Thanks for ref commernts it is easy tounderstan

---

## Author Comment (AC3)

Reply to Ref #1

I thank the referee for his valuable comments to improve my manuscript.
Will first correct a mistake. 1) Ref. conclude: data used in not open to all, is not correct.
Sources for these data – where they are published, are listed in manuscript. Can also be
copied at National Archie, is Oslo, where the Gjøa data are stored.   At Royal Sciety, London
or Scott's Musum for Discovery data.
2) Ref. may not be familiar with 'polar cap disturbances >80º mag. lat,, the mag. K-index in
this region is nor correlate with Kp. Cf. e.g. papers – 40 yrs ago by Dr. T.N. Davies,
Fairbanks, Geophys. Inst. Her polar rain & photo-electrons more important than sw.

>   3)   I will not delete the lines 36-39 of the abstract, but rewrite as: The main aim of
>        the paper is to establish a relation between a few types of polar cap auroras and
>        geomagnetic signatures , solar UV& X activity.
.
2. I cannot see why my manuscript is poorly organized. Can you name deficiencies? The
other referee liked it.
3. I will include a sentence with information about the solar cycle, but I don't think
important. ' All data are collected during solar cycle 14. The sunspot number varied from 36
to 48, but 120 years ago the values may be uncertain.
4. I agree that Fig. 1 is not so easy to read in all details, but its purpose is just to show the
locations of our two observations are located at the opposite end of the same field line – and
therefore share the same MLT. Equal important. The 2 stations are separated in local solar
time by 6.5 hours; i.e. you can distinguish between variation due to solar wind and variation
due to solar UV & X radiations & photoelectrons. The author feels the fig is unique. You
know where to fine a beather Fig?
5. I think the lines 101-103 clarify the local times in terms of UT which is helpful and
repeated in Table 1. The stations GH and CA are unique, because they have same MLT, but
different local time by 6.5 hrs. Fig. 1 is unique – the 2 stations are at opposite end of same
field line.
6. Details of the re-calculation. Have used the data center advice in Tokyo.
http://wdc.kugi.kyoto-u.ac.jp/index.html (WDC for Geomagnetism, Kyoto (kyoto-u.ac.jp)
7. Could you include the magnetic latitude of the two stations? GH = 78º N and CA = 78.7º S.
8. I will include the station name in line 117. Answere: general def of mag time, Greenwitch
already mentioned.Don't understand why
9. Could you include a date in line 197?   Done
10. Could you give the requested details in Fig. 4? Based on the auroral rep in logbook and
Amundsen's description in diary – I tried to reproduce this event by drawing - my sketch and
explained how in fig. text. Nb. The auroral ring – the oval, was never reported more than 30
degrees over southern horizon.
11. Could you clarify this in line 222?   Done
12. Answere:   Redrawn the CA magnetogram to show more clearly the detailed variations in
the H-comp. Fig also referred to in Ch. 6. The purpose of Fig. 5 is to show the geomagnetic
conditions on 4. & 5. Nov. 1903.
13. line 254: Why do you know that the color was, deeply reddish"? . As mentioned in
logbook, auroral color listed 7 times in addition to what written in the dairies.
14. line 279: Could you clarify this? Answere: The observations show.
15.Table 2: Could you clarify this? Amnswere: copied the original from Graaud, 1932. From
detailed checking diaries, I found 16 more events, which have been included in the months
when they were obserced.

16.line 317: Can you give a source of the statement 1903 was special with the strongest.." 31 October 1903 the strongest magnetic that century (cf. Egeland and Deehr, 21014).
I will not delete the sentence 317-318 about the Halloween storm. Perhaps we observe a 100 yrs period. A 100 yrs pereiod in auroral data also reported by Sam Silverman. It is interesting

17. about streamer: Both in data from GH & CA streamers are used many times.
18. I cannot delete sections 6.1 - 6.3 which are my main findings. (Will not. As written in Abstract and other places. This is the main findings in this study)
19: Could you follow the referee and delete lines 530-535? No, cf. Aans:were to point 18.
20. The data are open.
21.I would recommend to delete the last sentence of the manuscript 555-557. No, That is what Fig 12 showa.
22. I am uncertain about Fig. 6. A very interested figure, detailed explained in the 2 references listed in the figure .text
23. Magnetic time is included in the Fig. 9. Answere: To illustrate not correlatied with MLT as observed in th oval
24. Could you comment on the last remark of the referee? If some wish to work further with Gjøahavn data, they will find some valuable info through this reference.

---

## Author Response (AR1)

**Combined response to the referees:**

First; I wish to thank the referees both for critical comments and some unexpected questions regarding conjugate auroras and plasma physics in my manuscript on the results of the120 yrs old conjugate, polar cap auroral data recorded during the Gjøa and Discovery expeditions!!
I have – to the best of my knowledge, answered your questions and comments rg. the content. I have even pointed out – what I believe are some mistakes/misunderstand in your referees. I agree that my English is far from excellent, but should be understandable.
I don't agree that Fig. 1 is not illustrative and shows some interesting effects. Simultaneous geographic and magnetic coordinates plotted on a sphere, is difficult to plot. Have looked around and not found a better figure to illustrate that the two stations are at opposite end of the same field line which connect to the Earth near 78º magnetic latitude.

Second: My main aim with this work was to check if also polar cap auroras are mainly a geomagnetic phenomenon or if solar e-m radiation is an important source for auroras at polar latitudes. In addition, if it is possible from magnetic signatures to conclude that there are auroras overhead, even if you don't see them.

I have tried hard in my last version to change and update the language, but without changing the physical results from this investigation. The text in chapters 6 & 7 is updated and adjusted, but the conclusions are not changed. Chapter 7.1 have been cut out.
Thanks for pointing out I had forgot a reference. The final reference list has been checked and updated.
An auroral observer get suspicious if auroras are not observed a couple of hours around magnetic midnight, or when the daily variations of auroras follow the daily magnetic Sq variation.

To your information, I have not changed the lay out of the paper and even kept the figures. However, to update my English, I have made a lot of small changes in the original text, both a few new sentences and changed several words. By these changes I hope the quality has increased.
March 2024, Alv Egeland.

**Changes in my last version of the updated manuscript on "conjugate auroras".**
My main aim with this work was to check if also polar cap auroras are mainly a geomagnetic phenomenon or if solar e-m radiation is an important source. In addition, if it is possible from magnetic signatures to conclude that there are auroras overhead, even if you don't see them.

The Science results are not changed in the last version, but I have tried hard to update my English so the reader clearly understand what I try to point out.
1. Rg. The lay out: Have cut out Ch. 7.1, but kept the other chapters and the figures.
2. Chs. 6 & 7. The physical characteristics, particular the test rg. the form called SAA have been rewritten – not changing the physics, in a more modern form, which I hope is more clear English.  I conclude stronger that this polar cap auroral form is likely generated by photoelectrons because they show a daily variation similar to the Sq curve of the daily magnetic records.
3. I have added 3 new references in the final list and in the test.
4. In the other chapters + Abstract, several small – a few sentences, but mainly several words, and language changes have been made to improve the English. As I have not stored the earlier version, I can unfortunately not be more detailed/accurate.

5. Still the main conclusion – the characteristics of polar cap auroras are different from those observed in the oval, as pointed ot in the first version.

I certainly hope this is what you need. If you have proposals and questions, I will try to answer quickly.

Best regards Alv.